# Is ABO-Incompatible Living Donor Liver Transplantation Really a Good Alternative for Pediatric Recipients?

**DOI:** 10.3390/children8070600

**Published:** 2021-07-16

**Authors:** Catherine de Magnée, Louise Brunée, Roberto Tambucci, Aurore Pire, Isabelle Scheers, Etienne M. Sokal, Pamela Baldin, Francis Zech, Stéphane Eeckhoudt, Raymond Reding, Xavier Stephenne

**Affiliations:** 1Pediatric Surgery and Transplantation Unit, Cliniques Universitaires St Luc, Avenue Hippocrate 10, 1200 Brussels, Belgium; louise.brunee@student.uclouvain.be (L.B.); roberto.tambucci@uclouvain.be (R.T.); aurore.pire@uclouvain.be (A.P.); raymond.reding@uclouvain.be (R.R.); 2Pediatric Gastroenterology and Hepatology Division, Cliniques Universitaires St Luc, 1200 Brussels, Belgium; isabelle.scheers@uclouvain.be (I.S.); etienne.sokal@uclouvain.be (E.M.S.); xavier.stephenne@uclouvain.be (X.S.); 3Pathology Department, Cliniques Universitaires St Luc, 1200 Brussels, Belgium; pamela.baldin@uclouvain.be; 4Institute of Experimental and Clinical Research, Université Catholique de Louvain, 1348 Brussels, Belgium; francis.zech@uclouvain.be; 5Laboratoire Hospitalier Universitaire de Bruxelles, Université Libre de Bruxelles, 1050 Brussels, Belgium; stephane.eeckhoudt@lhub-ulb.be

**Keywords:** pediatric liver transplantation, ABO incompatibility, biliary complications, acute humoral rejection, C4d immunostaining

## Abstract

Background: ABO-incompatible (ABOi) living donor liver transplantation (LDLT) has been proposed to compensate for donor shortage. To date, few studies have reported detailed ABOi LDLT results in large series of pediatric patients. C4d complement deposition in graft capillaries has been reported to be associated with antibody-mediated rejection in solid organ transplantation. Methods: A retrospective case–control study was conducted, comparing clinical outcomes of each of 34 consecutive pediatric ABOi LDLT recipients with those of 2 non-ABOi pairs (*n* = 68), matched according to pre-transplant diagnostic criteria, age, and date of transplantation. In addition, we studied the C4d immunostaining pattern in 22 ABOi and in 36 non-ABOi recipients whose liver biopsy was performed within the first 4 post-transplant weeks for suspected acute rejection. Results: The incidence of biliary complications was higher in ABOi recipients (*p* < 0.05), as were the incidence of acute humoral rejection (*p* < 0.01) and the incidence of retransplantation (*p* < 0.05). All children who required retransplantation were older than 1 year at the time of ABOi LDLT. Positive C4d immunostaining was observed in 13/22 (59%) ABOi recipients versus 3/36 (8.3%) non-ABOi recipients (*p* < 0.0001). Conclusions: ABOi LDLT is a feasible option for pediatric end-stage liver disease but carries increased risks for the recipient, especially for children older than 1 year, even with a specific preparation protocol. C4d immunostaining may be a hallmark of acute humoral rejection in ABOi liver transplantation.

## 1. Introduction

Orthotopic liver transplantation (LT) nowadays constitutes a validated treatment for acute liver failure, end-stage liver disease, liver tumors, and selected metabolic disorders in children. The high mortality on the waiting list, due to the shortage of size-matched postmortem grafts, has led to the development of alternative techniques. Thomas Starzl described the liver as “a privileged organ” as it resists acute rejection in experimental animal studies [1]. Starzl was therefore the first to describe 11 cases of ABO-incompatible (ABOi) pediatric LT in 1979 because of the difficulty in finding compatible small grafts. He did not describe any case of acute rejection in these patients [2].

In the 1980s and in the early 1990s, several groups subsequently published their results concerning ABOi LT but with poor outcomes. The main encountered complications with ABOi LT were hepatic artery thrombosis, severe rejection, and increased incidence of biliary complications, including diffuse intrahepatic bile duct injuries [3,4]. In ABOi LT, acute humoral rejection (AHR) triggered by antibodies against donor-type iso-agglutinins is the most serious form of rejection and is often associated with graft loss. C4d complement deposition in graft capillaries has been reported to be associated with antibody-mediated rejection in kidney and other solid organ transplantations [5,6].

To overcome complications related to ABOi LT, several protocols were proposed in the 1990s, including high-dose immunosuppression, splenectomy, and plasmapheresis, but they had little effect on poor outcomes and increased the occurrence of specific complications, including infections and sepsis [7,8]. The impact of preformed ABO antibodies against the donor and specific strategies to reduce their titers were then described by several centers in the 1990s as a good strategy to improve ABOi LT outcomes [8,9,10,11]. Subsequently, the introduction of anti-CD20 monoclonal antibody (rituximab) led to a significant improvement in the survival rates of ABOi pediatric and adult living donor liver transplantation (LDLT) [12,13,14,15,16].

The objective of this study was to retrospectively review a single pediatric LT center’s 14-year experience with ABOi pediatric LDLT to determine whether outcomes with ABOi donors are truly comparable to ABO-compatible (ABOc) LDLT. Moreover, the impact of recipient age at the time of ABOi LT on post-transplant complications was investigated as it was hypothesized that children younger than 1 year are immunotolerant. It was also hypothesized that positive C4d immunostaining is more frequently observed in ABOi recipients’ biopsies taken within the immediate post-transplant period and could be a hallmark of underestimated AHR in ABOi pediatric LDLT.

## 2. Materials and Methods

### 2.1. Study Population

The medical records of 34 consecutive pediatric patients (<18 years of age) who received a primary ABOi LT between February 2005 and April 2019 at the Cliniques Universitaires St Luc in Brussels, Belgium, were retrospectively reviewed. The median age at transplantation was 0.9 years (range 0.4–14) (17 boys and 17 girls). Pre-transplant diagnoses were biliary atresia (*n* = 24/34, 70.6%), liver tumor (*n* = 5, 14.7%), progressive familial intrahepatic cholestasis (*n* = 2, 5.9%), Alagille’s syndrome (*n* = 1, 2.9%), metabolic disease (*n* = 1, 2.9%), and others (*n* = 1, 2.9%). Eighteen patients (52.9%) were under 1 year of age at the time of transplantation. All patients received a living related liver graft. The minimal follow-up period was 1 year for all children (mean follow-up: 5.7 ± 0.6 years).

For each ABOi LDLT case, 2 controls were selected from our overall non-ABOi LDLT series, according to the following pairing criteria: (1) same pre-transplant diagnosis, (2) LT performed within 2 years before and after, and (3) the closest age at the time of LT. Sixty-eight pediatric patients (<18 years) then constituted the control group. They received a primary ABOc LT between September 2003 and August 2019 at the Cliniques Universitaires St Luc in Brussels, Belgium. The median age at transplantation was 1 year (range 0.5–14.3) (32 boys and 36 girls). Pre-transplant diagnoses were as follows: biliary atresia (*n* = 48/68, 70.6%), liver tumor (*n* = 10, 14.7%), progressive familial intrahepatic cholestasis (*n* = 4, 5.9%), Alagille’s syndrome (*n* = 2, 2.9%), metabolic disease (*n* = 2, 2.9%), and others (*n* = 2, 2.9%). Thirty-four patients (50%) were under 1 year of age at the time of transplantation. All patients received a living related liver graft. The minimal follow-up was 1 year for all children (mean follow-up: 5.5 ± 0.5 years).

Technical details regarding donor surgery, LT procedures, and postoperative management have been previously described [17,18,19,20,21].

The study was approved by the hospital ethics committee (2019/11JUI/247; approval number B 403).

### 2.2. Basic Immunosuppressive Protocol

Primary baseline immunosuppression for all LT recipients consisted of a bi-therapy including a calcineurin inhibitor (tacrolimus; Prograft^®^; Astellas, Tokyo, Japan) and a monoclonal anti-CD 25 antibody (basiliximab; Simulect^®^; Novartis, Bâle, Switzerland). We applied a basic immunosuppressive protocol without steroids [22]. Basiliximab was administered for intravenous induction on day 0 and day 4 after LT, the dose being adapted based on the recipient’s body weight (<15 kg, 5 mg; 15–25 kg, 10 mg; >25 kg, 20 mg). Patients received oral tacrolimus from the day of transplantation, with an initial dose of 0.1 mg/kg/12 h. Target tacrolimus levels varied according to each patient’s clinical condition but were generally 8–10 ng/mL during the first 2 months after LT, 6–8 ng/mL during the third month after transplantation, and 4–6 ng/mL between 3 months and 1 year post-transplantation. When acute cellular rejection (ACR) was suspected on the basis of abnormal liver function tests and/or on clinical parameters (unexplained fever, severe unexplained ascites, etc.), a liver biopsy was performed. Proven ACR on liver biopsy (according to BANFF classification [22]) was treated with high-dose methylprednisolone (Solumedrol^®^, Medrol^®^; Pfizer, New York, NY, USA): 5 mg/kg for 3 days, 3.75/kg mg for 1 day, 2.5 mg/kg for 1 day, and 1.25 mg/kg for 1 day to maintain a dose of 1 mg/kg for 1 week. Depending on clinical and biological evolutions, patients were switched to oral steroid therapy, and steroid dosage was subsequently tapered to reach 0.25 mg/kg/day at 3 months post-LT and 0.5 mg/kg/2 days until the systematic post-LT biopsy at 6 months.

### 2.3. Immunosuppressive Protocol for ABOi Pediatric LDLT

We performed a total of 34 ABOi pediatric LDLTs, in which 18 patients (52.9%) were less than 1 year old at the time of transplantation. Anti-donor blood group antibodies were systematically monitored in all candidate ABOi recipients during the pre-transplant workup. Natural anti-ABO titers were measured using a simple serial test tube dilution method. For the immune part of the anti-A and/or anti-B antibodies, a column agglutination technique was used after hydrolysis of the patients’ plasma with dithiothreitol. If natural and/or immune iso-agglutinin titers were ≥1/16 at pre-transplant workup, a specific preparation protocol was applied: 375 mg/m^2^ rituximab (Rituxan^®^; Mabthera, Roche, Welwyn Garden City, UK) on day 14 before LT. In patients with anti-donor blood group antibodies ≥1/16 on day 7 before transplantation, plasmapheresis was performed as well. If iso-agglutinins remained at ≥1/16 on day 5 before transplantation, patients received both a further plasmapheresis and an immunoglobulin injection (sandoglobuline^®^; CSL Behring, Pennsylvania, PA, USA) (100 mg/kg). Additional plasmaphereses were done on day 1 before LT if iso-agglutinins remained at ≥1/16 despite the previous preparation protocol. All ABOi patients also received the baseline immunosuppressive protocol described above. The target level of tacrolimus was the same as in non-ABOi cases. For ABOi LDLT cases, steroids were added to immunosuppressive therapy and initiated with an injection of 2 mg/kg of methylprednisolone before graft perfusion during surgery. Recipients received an intravenous injection of 1 mg/kg of methylprednisolone during postoperative day (POD) 1–2, 0.75 mg/kg during POD 3–5, 0.5 mg/kg during POD 6–10, 0.4 mg/kg during POD 11–15, 0.3 mg/kg during POD 16–20, and 0.25 mg/kg during POD 25–50. Patients were switched to oral methylprednisolone as soon as possible, and steroid dosage was subsequently tapered (as previously explained) until the result of the routine post-transplant liver biopsy at 6 months (depending on the results of the latter).

### 2.4. Monitoring of Iso-Agglutinins in the Post-Transplant Period for ABOi Recipients and Adjustment of Immunosuppressive Therapy

Natural and immune iso-agglutinins were monitored daily for the first 8 days after ABOi LT and then at POD 10, 15, 20, and 30 (using the technique previously described). Additional dosages of immune and natural iso-agglutinins were performed depending on the clinical and biological course of each patient. Depending on the post-LT iso-agglutinin titers, additional immunosuppressive treatments were applied: plasmapheresis, rituximab, and/or immunoglobulins.

### 2.5. Histopathological Studies on Post-LT Liver Biopsies for Suspected ACR

Pathological diagnosis was made on a routine basis by 3 pathologists. The minimal quantitative requirement for the diagnosis of rejection was a biopsy containing at least 5 portal spaces. Liver biopsies were formalin-fixed (10%), paraffin-embedded, and cut into 4 µm sections. Staining methods for routine histological evaluation included hematoxylin and eosin, Period acid-Shiff-diastase (PAS-D), Masson’s trichrome, iron, and immunostaining for cytokeratin 7.

When acute rejection was suspected based on abnormal liver function tests and/or on clinical parameters (unexplained fever, severe unexplained ascites, etc.), a liver biopsy was performed. Twenty-two (64.7%) ABOi recipients and thirty-six (52.9%) ABOc recipients underwent a liver biopsy according to these criteria within the first 4 weeks post-transplant. ACR was diagnosed using Banff criteria [22]. Histological AHR was suspected when peri-portal edema and necrosis, or portal hemorrhagic edema was associated with elevated antidonor A/B antibody titers [23,24].

In each case of ACR and AHR, portal inflammation, bile duct damage, and venular endothelialitis were assessed separately using the rejection activity index of Banff criteria [22]. Immunostaining for C4d (rabbit monoclonal SP91, abcam, Cambridge, UK) was additionally performed, but AHR definition was not based on this C4d immunostaining. We considered the immunostaining for C4d positive when endothelial cells of portal veins and/or portal capillaries were strongly positive [25].

## 3. Statistical Analysis

Cases and controls were verified to be correctly matched for pre-transplant diagnosis, age, and date of transplantation. Continuous variables were normalized by Box and Cox transformation. Comparisons of relevant variables between cases and donors were assessed by mixed model regressions, treating the average outcomes of each set of matched patients as a random variable. These regressions were performed according to a mixed linear, logistic, or Weibull model, depending on the nature of the dependent variable. Post-transplant liver biopsy (performed in the case of suspected acute rejection) was not available for all patients, and we had to rely on classical tests rather than mixed models because of the lack of match between the two groups. The Banff score being an ordered variable, we used a regression according to Snell [26]. Comparisons of variables between two age groups were made by Student’s *t*-test, Weibull model regression, or the chi-square test (with Cook’s correction), depending on the nature of the variable studied. When the numbers studied did not meet the criteria for a chi-square, Fisher’s exact test was used. To minimize the bias related to the asymmetry of the independent variables, we used a sandwich variance for the regressions and calculated the degrees of freedom according to Bell and McCaffrey [27]. Statistical significance was set at two-sided * *p* < 0.05, ** *p* < 0.01, and *** *p* < 0.001.

## 4. Results

### 4.1. Overall Results

Characteristics of ABOi pediatric liver recipients (*n* = 34) and their control group of ABOc LDLT (*n* = 68) are summarized in Table 1.

The 2-year patient and graft survival rates for ABOi LDLT recipients were 100% and 91.2%, respectively. For ABOc LDLT recipients, the 2-year patient and graft survival rates were 95.6% and 95.6%, respectively. The 5-year patient and graft survival rates remained unchanged in both groups. Three ABOc recipients died during the follow-up period: 1 patient because of sepsis and 2 children transplanted for liver malignancies in the setting of post-LT metastatic disease. Three ABOi recipients required retransplantation for diffuse intrahepatic bile duct injuries.

### 4.2. Pre-Transplant Iso-Agglutinins and Immunosuppressive Preparation in ABOi LDLT Recipients

Among 34 ABOi LDLT recipients, 10 (29.4%) had immune and/or natural iso-agglutinin titers of ≥1/16 at pre-LT workup. According to these results and our immunosuppressive preparation for ABOi LT, 10 children (29.4%) received rituximab at day 14 before LT. After this initial immunological preparation, 8/34 patients (23.5%) required plasmapheresis in the pre-LT period (between one and three sessions) and 1/34 (2.9%) required immunoglobulins.

Among patients <1 year of age at transplantation (*n* = 18), 2/18 (11.1%) had immune and/or natural iso-agglutinin titers of ≥1/16 before LT, 2/18 (11.1%) required rituximab at day 14 before LT, 1/18 (5.5%) required plasmapheresis (one session), but none required immunoglobulins.

Among patients ≥1 year of age at the time of transplantation (*n* = 16), 8/16 (50%) had immune and/or natural iso-agglutinin titers of ≥1/16 at pre-LT workup, 8 (50%) received rituximab at day 14 before LT, 7/16 (43.7%) plasmapheresis (between one and three sessions), and 1/16 (6.2%) immunoglobulins.

Children older than 1 year required a significantly heavier immunosuppressive preparation before ABOi LT. Indeed, they more frequently needed rituximab administration (HR = 7.5 (95% CI: 1.13–88.7); *p* = 0.02) and plasmapheresis (HR = 12.2 (95% CI: 1.3–627.5); *p* = 0.01) compared with children younger than 1 year.

Positive correlations were observed between both natural and immune iso-agglutinins at pre-LT workup and the recipient’s age at transplantation (*p* = 0.001 and *p* < 0.01, respectively) (Figure 1).

### 4.3. Post-Transplant Iso-Agglutinins and Additional Post-LT Immunosuppressive Treatments in ABOi LDLT Recipients

Among 34 pediatric recipients of an ABOi LDLT, 14/34 (41.1%) had immune and/or natural iso-agglutinin titers of ≥1/16 between day 1 and day 30 post-LT. According to these results and our immunosuppressive protocol for ABOi LT, 14/34 (41.1%) children received rituximab, 6/34 (17.6%) plasmapheresis (between two and six sessions), and 5/34 (14.7%) immunoglobulins in the early post-LT period. Of these 14 patients with iso-agglutinin titers of ≥1/16 during the first month post-LT, 10 (71.4%) already had high titers of iso-agglutinins (≥1/16) at pre-LT workup (and were submitted to our specific ABOi pre-LT immunosuppressive preparation).

Among patients <1 year of age at the time of transplantation (*n* = 18), 1/18 (5.5%) had immune and/or natural iso-agglutinin titers of ≥1/16 at POD 1–30, 1/18 (5.5%) required rituximab, 1/18 (5.5%) plasmapheresis (two sessions), and 1/18 (5.5%) immunoglobulins.

Among patients ≥1 year of age at transplantation (*n* = 16), 13/16 (81.2%) had immune and/or natural iso-agglutinin titers of ≥1/16 between day 1 and 30 post-LT, 13/16 (81.2%) received rituximab, 5/16 (31.2%) plasmapheresis (between two and six sessions), and 4/16 (25%) immunoglobulins in the early post-LT period.

Children older than 1 year required higher immunosuppression after ABOi LT. Indeed, they more frequently required rituximab administration (HR = 59.2 (95% CI: 5.7–3218.2); *p* < 0.0001) and plasmapheresis (HR = 7.3 (95% CI: 0.7–385.5); *p* = 0.08 NS) when compared with those younger than 1 year old.

Positive correlations were found between the maximum titers of post-LT natural and immune iso-agglutinins and the recipient’s age at transplantation (*p* < 0.01 and *p* < 0.05; respectively) (Figure 2).

### 4.4. Post-LT Complications

Among the 102 patients, 3 (2.9%) required retransplantation for diffuse intrahepatic bile duct injuries. Six children (5.9%) developed hepatic artery thrombosis, 8 (7.8%) a portal vein complication; 2 (2%), Budd–Chiari syndrome or outflow obstruction; and 24 (23.5%), a biliary complication (biliary stricture or fistula) in the post-LT period. Regarding immunological complications, 64 patients (62.7%) developed ACR treated by steroids, 7 children (6.9%) had corticosteroid-resistant ACR, 4 children (3.9%) had AHR, no patients had chronic rejection, and 7 patients (6.9%) developed a post-LT lymphoproliferative disease.

The incidences of major post-LT complications according to ABO compatibility between donor and recipient are presented in Table 2.

The incidences of major post-ABOi LT complications according to recipient age (<1 year versus ≥1 year) are summarized in Table 3.

As shown in Table 3, all ABOi LT recipients who required retransplantation were older than 1 year at the time of transplantation. The indication for retransplantation was diffuse intrahepatic bile duct injuries in all cases, an entity that always occurred within the first 4 months after ABOi LT.

A positive correlation was observed between immediate pre-transplant immune iso-agglutinin titers (after the immunosuppressive preparation applied for ABOi LDLT) and the incidence of post-LT AHR (*p* < 0.05). A positive correlation was also observed between the maximum post-transplant immune iso-agglutinin titer and the incidence of several post-LT complications: ACR (*p* = 0.01), AHR (*p* < 0.01), and retransplantation (*p* < 0.01). Similarly, a positive correlation was observed between the maximum post-transplant natural iso-agglutinin titer and AHR (*p* < 0.05) and retransplantation (*p* < 0.05).

### 4.5. Histopathologic Studies on Post-Transplant Liver Biopsies

Among the 22 ABOi and 36 ABOc recipients who underwent a liver biopsy during the first 4 weeks post-LT for suspected ACR, the Banff score was not significantly different between the two groups of patients (median Banff for ABOi patients = 5; median BANFF for ABOc recipients = 6; *p* = 0.8 NS).

On these liver biopsies taken for suspected ACR, positive C4d immunostaining was observed in 13/22 (59%) ABOi recipients versus in 3/36 (8.3%) non-ABOi recipients (*p* < 0.0001) (Figure 3).

All ABOi recipients who required retransplantation had positive C4d immunostaining on their liver graft biopsy. We also observed that ABOi recipients with positive C4d immunostaining had a significantly higher maximum post-LT immune iso-agglutinin titer (*p* < 0.05) and required plasmapheresis more frequently in the post-transplant period compared with ABOi C4d-negative patients (*p* < 0.05).

## 5. Discussion

According to our results, (1) despite a specific immunosuppressive protocol, the incidence of biliary complications was higher in ABOi pediatric LT recipients, as was the incidence of AHR, and the incidence of retransplantation; (2) all children who required retransplantation (for diffuse bile duct injuries related to ABO incompatibility) were older than 1 year of age at the time of ABOi LDLT; and (3) liver biopsies performed early after transplantation revealed positive C4d immunostaining mainly in ABOi recipients.

Despite Starzl’s description of the liver as “a privileged organ”, outcomes of the first studies including ABOi LT were poor. For this reason, ABOi liver grafts were initially limited to emergency situations. Indeed, Gordon et al. reported decreased patient (45%) and graft (26%) survival rates for ABOi LT at 1 year post-transplantation [28]. Similarly, in 1990, Gugenheim et al. published a series of 234 LT cases, in which the 2-year graft survival rates were 76% for ABOc and 30% for ABOi LT [7]. They noted an increased incidence of biliary complications, vascular thrombosis, cellular rejection, and histological findings consistent with antibody-mediated rejection. ABO antigens are expressed on the surface of vascular endothelial cells, bile duct cells, and sinusoidal endothelial cells of the transplanted liver. During ABOi LT, anti-blood-type antibodies bind to corresponding antigens on the surface of vascular endothelial cells and form antigen–antibody complexes that activate the complement system, resulting in complications, such as acute rejection and biliary complications, eventually leading to graft failure [4]. More recently, several groups have described new immunosuppressive strategies to improve graft survival during blood group cross-over: plasmapheresis, splenectomy, intravenous immunoglobulins, and arterial infusion therapy; but these strategies have little effect on poor outcomes and have increased the occurrence of other complications, including infections and sepsis [7,8,29,30]. A key element in ABOi LT was the description of the first case of ABOi LT using rituximab (a chimeric human-murine anti-CD20 monoclonal antibody) in 2003 and the first case of rituximab prophylaxis in ABOi LDLT in 2005 [30,31]. Indeed, the outcomes of ABOi LDLT have been radically improved since the introduction of this therapy. Egawa et al. published a large series including 291 ABOi LDLT cases in Japan, with a 3-year survival rate that increased from 30% to 80% after the introduction of rituximab [32,33]. Pediatric recipients tend to have better postoperative outcomes than adults after ABOi LT, because of the low levels of anti-ABO antibody titers in children, which stem from their immature immune and complement systems. However, outcomes of ABOi LT in pediatric populations remain uncertain and controversial.

According to our results, the incidences of biliary complications, AHR, and retransplantation were significantly higher in ABOi LDLT recipients despite a specific immunosuppressive pre- and post-transplant protocol. In 1994, Tanaka et al. reported 13 cases of ABOi pediatric LDLT, with 77% 1-year patient and graft survival rates [34]. Another study published in 1999 by Varela-Fascinetto et al. observed no significant difference in 10-year patient or graft survival rates in children who received 28 ABOi grafts compared with those who received 72 ABOc grafts [35]. In contrast, Szymczak et al. reported a 57% decrease in 1-year patient and graft survival rates in children, associated with increased acute and chronic rejection of ABOi grafts [36].

It has been suggested that more acceptable ABOi LT outcomes may be achieved in younger children [37,38]. Yandza et al. demonstrated that children under 2 years of age have lower anti-ABO antibody titers and lower morbidity than adults [10]. Gelas et al. found that among infants less than 1 year old, outcomes of ABOi LT are similar to those from ABOc LT, whereas such comparability was not observed in older children [39]. Similarly, Egawa et al. showed that the 5-year patient survival rates in patients younger than 1 year and older than 16 years are 77% and 22%, respectively. They observed that the incidence of intrahepatic biliary complications and hepatic necrosis in ABOi LDLT increases significantly with recipient age. Predictive risk factors for increased morbidity and mortality in this study were age ≥ 1 year and high anti-ABO antibody titers before transplantation [40]. These observations are consistent with our results as we described: (1) positive correlations between pre- and post-transplant immune and natural iso-agglutinins and the age of the recipient at the time of transplantation and (2) positive correlations between pre- and post-transplant immune and natural iso-agglutinin titers and the incidence of several post-LT complications: ACR, AHR, and retransplantation. Indeed, according to our results, children older than 1 year have higher titers of anti-donor blood group antibodies and require heavier immunosuppression protocols before and after ABOi LDLT. Moreover, an interesting finding is that all children who required retransplantation (due to diffuse intrahepatic bile duct injuries) in our series were older than 1 year. We can understand these results if we analyze anti-ABO antibody kinetics during childhood: maternal antibodies disappear from the neonate after 2 weeks, the newborn starts to produce its own IgM and IgG between 8 and 12 weeks after birth, and adult levels are reached around 5 to 10 years of age [41]. Other authors have also studied anti-ABO antibody titers before and after ABOi LDLT, describing (1) high pre-transplant anti-ABO antibody titers in patients ≥1 year of age; (2) a potential to increase anti-ABO antibody titers after LT, even if preoperative titers are low; and (3) a correlation between high antibody titers and severe post-LT complications (i.e., hepatic necrosis and intrahepatic biliary complications) [40,42]. In view of our results, as well as those of other authors [40], it may be questionable whether children older than 1 year of age are good candidates for ABOi LDLT.

In ABOi LT, AHR triggered by antibodies against donor-type iso-agglutinins is the most serious form of rejection and is often associated with graft loss. C4d is a molecule that binds to tissues after complement system activation, and immunostaining has been widely used to demonstrate humoral immunoreactivity or antibody-mediated rejection in kidney and heart transplantation [43,44]. Other authors have also demonstrated that C4d deposits in the portal stroma may be a feature of AHR in ABOi LT [45]. However, few authors have studied C4d status on liver biopsies taken for suspected acute rejection in both ABOi and ABOc pediatric LDLT recipients. When we analyzed C4d immunostaining on liver biopsies taken for suspected ACR within the first 4 weeks after LDLT, C4d-positive immunostaining was observed in 59% of ABOi recipients compared with 8.3% of non-ABOi recipients (*p* < 0.0001). We also observed that all ABOi recipients who required retransplantation for diffuse intrahepatic bile duct injuries had C4d-positive immunostaining on their post-transplant liver graft biopsy. Our results may suggest that limits between histologic ACR and AHR may be blurred and that C4d immunostaining may be routinely used in cases of suspected ACR in ABOi LT to guide the diagnosis toward underestimated AHR.

Interesting future work could consist of C4d immunostaining studies on systematic liver biopsies performed 6 months and 1 year after transplantation and their correlation with the medium-term clinical and biological course in ABOi versus ABOc pediatric LDLT recipients.

## 6. Conclusions

ABO-incompatible LDLT is a feasible option for end-stage pediatric liver disease but entails increased risks for the recipient, with a higher incidence of biliary complications, AHR, and retransplantation despite a specific pre- and post-transplant immunosuppressive protocol. We need to be particularly careful with children older than 1 year as they have higher anti-donor blood group antibody titers, require heavier immunosuppressive protocols before and after ABOi LDLT, and have a higher rate of retransplantation. C4d immunostaining may be used routinely for suspected ACR in ABOi LT and may be a hallmark of underestimated AHR.

## Figures and Tables

**Figure 1 children-08-00600-f001:**
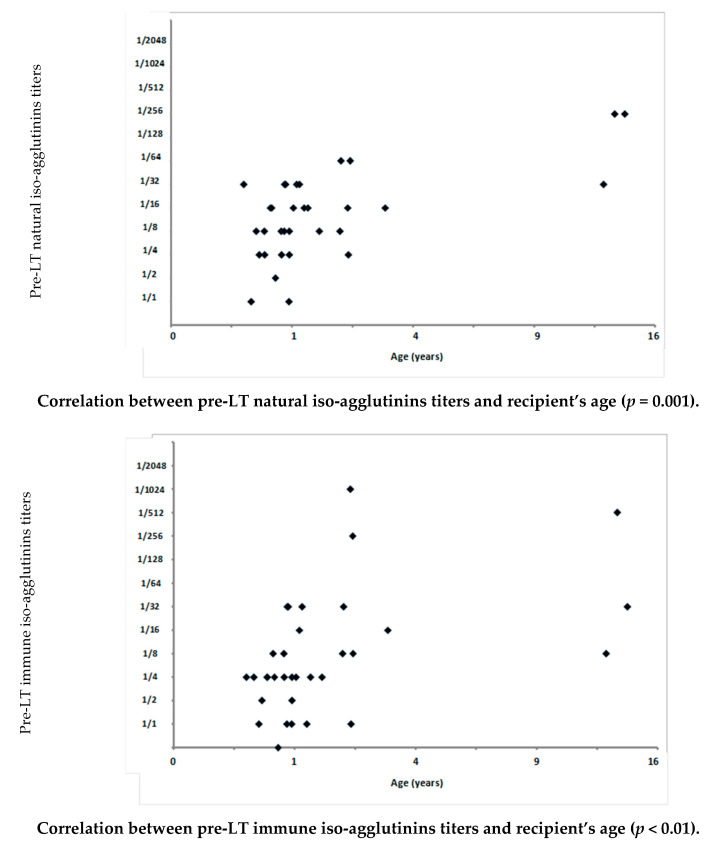
Correlations between pre-transplant anti-donor blood group antibodies (at pre-transplant workup) and the recipient’s age at the time of living donor liver transplantation performed in a series of 34 ABO-incompatible pediatric recipients between February 2005 and April 2019 at the Cliniques Universitaires St Luc in Brussels, Belgium. LT: liver transplantation.

**Figure 2 children-08-00600-f002:**
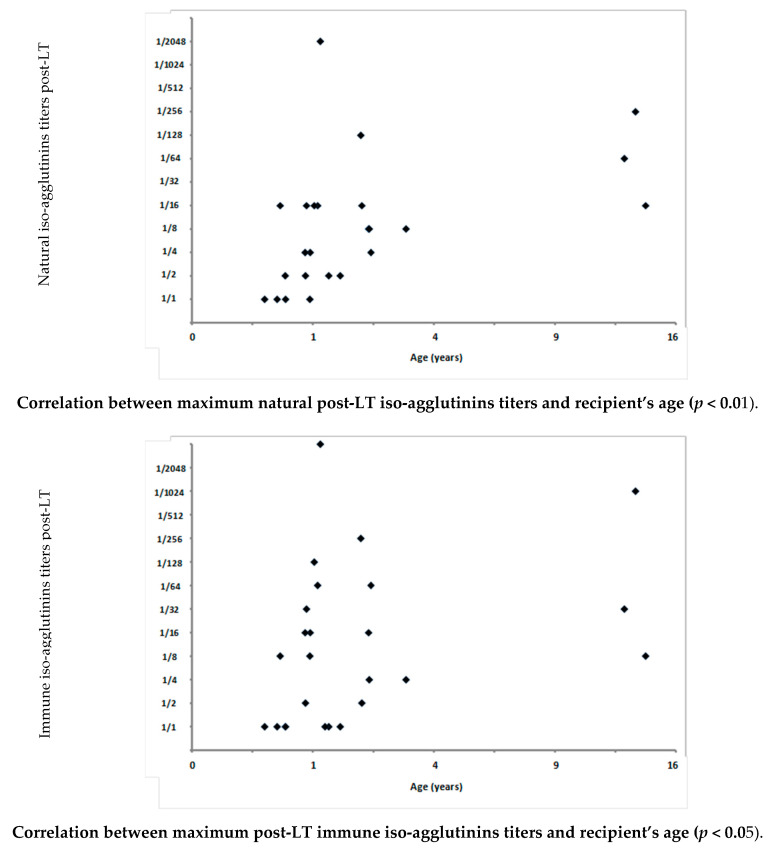
Correlations between maximum post-transplant anti-donor blood group antibodies and the recipient’s age at the time of living donor liver transplantation performed in a series of 34 ABO-incompatible recipients between February 2005 and April 2019 at the Cliniques Universitaires St Luc in Brussels, Belgium. LT: liver transplantation.

**Figure 3 children-08-00600-f003:**
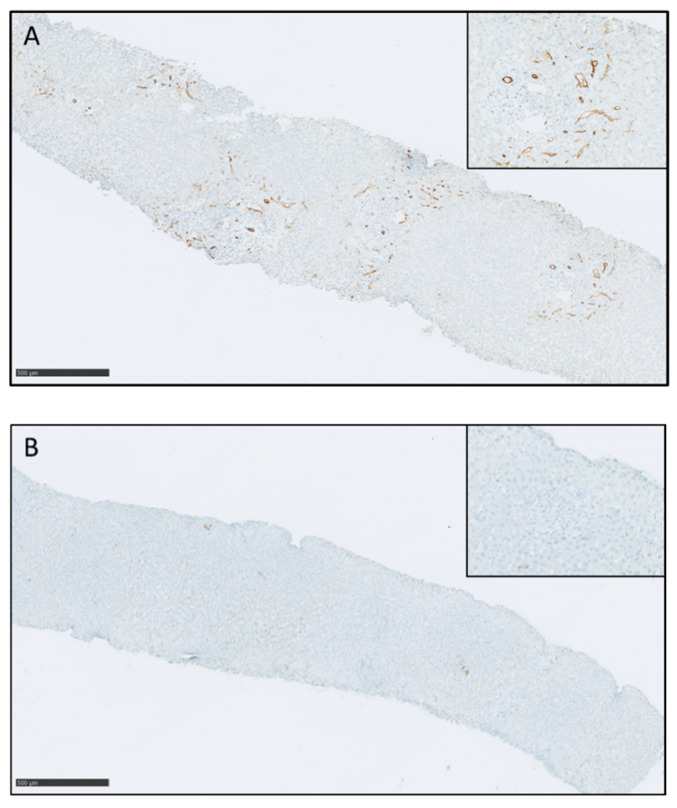
Immunostaining for C4d. Positivity of C4d staining in the vessels of the portal tracts ((**A**), magnification 5×); in the inset is shown the strong positivity of the staining in portal vessels and capillaries (arrow, magnification 20×). In (**B**), the staining is negative in all the biopsy (magnification 5×); in the inset, a completely negative portal tract (magnification 20×).

**Table 1 children-08-00600-t001:** Characteristics of 34 ABOi pediatric liver recipients and their control group of 68 ABOc living donor liver transplantations performed at the Cliniques Universitaires St Luc, Brussels, Belgium (September 2003–August 2019).

	ABOi LDLT (*n* = 34)	ABOc LDLT (*n* = 68)	*p*-Value
Transplantation date	+129 days of median delay (compared to ABOc LDLT)		0.18 (NS)
Recipient sex	17 girls/17 boys	36 girls/32 boys	0.79 (NS)
Median recipient age at transplantation (years)	0.95	0.98	0.10 (NS)
Median recipient weight (kg)	8.7	8	0.32 (NS)
Median recipient height (cm)	71.7	70.7	0.98 (NS)
Median recipient PELD	17	20	0.25 (NS)
Median donor age (years)	31	34	0.28 (NS)
Median ischemia time (min)	141	151	0.58 (NS)
Median graft weight/recipient weight (%)	2.9	3.1	0.50 (NS)

ABOi: ABO incompatible; ABOc: ABO compatible; LDLT: living donor liver transplantation; PELD: pediatric end-stage liver-disease score.

**Table 2 children-08-00600-t002:** Post-transplant complications according to ABO compatibility between donor and recipient among 34 ABOi and 68 non-ABOi pediatric liver transplantations performed at the Cliniques Universitaires St Luc, Brussels, Belgium (September 2003–August 2019).

	ABOi LT (*n* = 34)	Matched ABOc LT (*n* = 68)	Hazard Ratio ABOi/ABOc (95% CI)	*p*-Value
Hepatic artery complications	2 (5.9%)	4 (5.9%)	1 (0.2–4.6)	NS
Portal vein complications	3 (8.8%)	5 (7.3%)	1.2 (0.2–6.1)	NS
Budd–Chiari syndrome	1 (2.9%)	1 (1.5%)	1.3 (0.1–13.8)	NS
**Biliary complications**	13 (38.2%)	11 (16.2%)	3.2 (1.2–8.7)	<0.05
Acute cellular rejection	22 (64.7%)	34 (50%)	1.7 (0.7–4)	NS
Acute cellular rejection steroid resistant	4 (11.8%)	3 (4.4%)	2.9 (0.5–15.6)	NS
**Acute humoral rejection**	4 (11.8%)	0 (0%)	-	<0.01
Chronic rejection	0 (0%)	0 (0%)	-	NS
Post-LT lymphoproliferative disease	3 (8.8%)	4 (5.9%)	1.5 (0.3–6.7)	NS
**Retransplantation**	3 (8.8%)	0 (0%)	14 (1.5–132.9)	<0.05
Death	0 (0%)	3 (4.4%)	0.3 (0.02–4.6)	NS

ABOi: ABO incompatible; LT: liver transplantation; ABOc: ABO compatible; CI: confidence interval.

**Table 3 children-08-00600-t003:** Post-transplant complications according to recipient age (<1 year versus ≥1 year) among 34 ABO-incompatible liver transplantations performed at the Cliniques Universitaires St Luc (Brussels, Belgium) between February 2005 and April 2019.

	ABOi LT Recipients <1 Year Old (*n* = 18)	ABOi LT Recipients ≥1 Year Old (*n* = 16)	Hazard Ratio Children ≥1 Year/Children <1 Year (95% CI)	*p*-Value
Hepatic artery complications	1 (5.5%)	1 (6.2%)	1.1 (0.01–94)	NS
Portal vein complications	2 (11.1%)	1 (6.2%)	0.5 (0.008–11.4)	NS
Budd–Chiari syndrome	0 (0%)	1 (6.2%)	-	NS
Biliary complications	5 (27.8%)	8 (50%)	2.5 (0.5–13.8)	NS
Acute cellular rejection	13 (72.2%)	11 (68.7%)	0.8 (0.1–4.8)	NS
Acute cellular rejection steroid resistant	2 (11.1%)	2 (12.5%)	1.1 (0.07–17.6)	NS
Acute humoral rejection	1 (5.5%)	3 (18.7%)	3.7 (0.3–217.5)	NS
Chronic rejection	0 (0%)	0 (0%)	-	NS
Post-LT lymphoproliferative disease	2 (11.1%)	1 (6.2%)	0.5 (0.008–11.5)	NS
**Retransplantation**	0 (0%)	3 (18.7%)	-	NS (*p* = 0.09)
Death	0 (0%)	0 (0%)	-	NS

ABOi: ABO incompatible; LT: liver transplantation; ABOc: ABO compatible; CI confidence interval.

## Data Availability

Data can be provided by the first author: Catherine de Magnée: catherine.demagnee@uclouvain.be.

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
