# Peer review of "Is ABO-Incompatible Living Donor Liver Transplantation Really a Good Alternative for Pediatric Recipients?"

_children, 2021, doi:10.3390/children8070600_

Round 1

Reviewer 1 Report

It was very interesting to focus on age and examine the prognosis of transplantation. It seems to be a treatise mainly related to clinical findings. In conclusion, the significance of immunostaining is mentioned, but I think it is insufficiently explained and summarized. Although there is a description of the method of immunostaining itself, I think that the evaluation method and the actual tissue figures should be described in more detail. To prove the conclusions the authurs made, it would be better to explain the breakdown of the C4d positive and negative groups, a comparison of clinical findings, and histological findings. The author's contribution is not listed. This needs to be improved.

Author Response

Reviewer 1 mentioned that the author's contributions was not listed. But I did it in details during the submission process. 

All the reviewer's remarks and comments concerning C4d immunostaining were added to the article. Please see the attachment. The changes are underlined in the new document. We added details concerning C4d methodology (with a new bibliographic reference), figures illustrating the C4d positive and negative immunostaining (figure 3), and several correlations between C4d immunostaining and clinical evolution of the patients. 

However, if we consider ABOi recipients (n=34), 22 had a liver biopsy during the first 4 weeks post-transplant for suspected acute rejection. Among these 22 patients, 13 had a postive C4d immunostaining, and 9 had a negative C4d immunostaining. With these small numbers of patients in each group, it's difficult to have statistically significant results. 

Reviewer 2 Report

The authors present interesting data on the outcome of ABO-incompatible living donor liver transplantation (ABOi LDLT). The manuscript is well written, the study well designed, and the conclusions of this study are well supported by the data presented.

The paper brings important conclusions on the increased risk for biliary complications, acute humoral rejection, and incidence of transplantation in children with ABO-incompatible LDLT, despite the improved possibilities on the specific immunosuppressive protocol. Also, an important practical conclusion and possible idea for future work was the importance of analyzing C4d immunostaining on post-transplant liver biopsies to evaluate the risk for acute humoral rejection in ABOi LDLT.

Just a minor correction in Table 1. As the controls are chosen to match the diagnosis of the ABOi LDLT patients exactly, there is no need to include in the table also the pretransplant diagnosis (second row).

Author Response

Reviewer 2 asked to correct Table 1 (removing the second row). We did it and the Table 1 was corrected on the original article.

He also required minor English language and style revisions. We did it and included these revisions to the main text.

Round 2

Reviewer 1 Report

I saw the attached figures. It does not seem to be stained under the same conditions. The nuclear staining of two figures are too different. Unfortunately, the results do not seem to have been fully considered.

In addition, when making comparisons, it is desirable to use images of the same magnification.

Author Response

According to reviewer's comments, we completely changed Figure 3. 

Supplementary statistics were done studying correlations between C4d immunostaining and biological and clinical patient’s evolution. However, if we consider ABOi recipients (n=34), 22 had a liver biopsy during the first 4 weeks post-transplant for suspected acute rejection. Among these 22 patients, 13 had a positive C4d immunostaining, and 9 had a negative C4d immunostaining. With these small numbers of patients in each group, it's difficult to have statistically significant results.